# Research and Evaluation of the Influence of the Construction of the Gate and the Influence of the Piston Velocity on the Distribution of Gases into the Volume of the Casting

**DOI:** 10.3390/ma14092264

**Published:** 2021-04-27

**Authors:** Ján Majerník, Štefan Gašpár, Jozef Husár, Ján Paško, Jan Kolínský

**Affiliations:** 1Department of Mechanical Engineering, Faculty of Technology, Institute of Technology and Business in České Budějovice, Okružní 517/10, 370 01 České Budějovice, Czech Republic; majernik@mail.vstecb.cz (J.M.); kolinsky@mail.vstecb.cz (J.K.); 2Department of Technical Systems Design and Monitoring, Faculty of Manufacturing Technologies with a Seat in Prešov, Technical University of Košice, Štúrova 31, 08001 Prešov, Slovakia; stefan.gaspar@tuke.sk (Š.G.); jan.pasko@tuke.sk (J.P.); 3Department of Industrial Engineering and Informatics, Faculty of Manufacturing Technologies with a Seat in Prešov, Technical University of Košice, Bayerova 1, 08001 Prešov, Slovakia

**Keywords:** HPDC, technological factors of die casting, gating system, aluminum alloys, air entrapment

## Abstract

Distribution of gasses to the cast volume and volume of pores can be maintained within the acceptable limits by means of correct setting of technological parameters of casting and by selection of suitable structure and gating system arrangement. The main idea of this paper solves the issue of suitability of die casting adjustment—i.e., change of technological parameters or change of structural solution of the gating system—with regards to inner soundness of casts produced in die casting process. Parameters which were compared included height of a gate and velocity of a piston. The melt velocity in the gate was used as a correlating factor between the gate height and piston velocity. The evaluated parameter was gas entrapment in the cast at the end of the filling phase of die casting cycle and at the same time percentage of porosity in the samples taken from the main runner. On the basis of the performed experiments it was proved that the change of technological parameters, particularly of pressing velocity of the piston, directly influences distribution of gasses to the cast volume.

## 1. Introduction

High pressure die casting (HPDC) allows production of thin-walled casts with high geometrical precision, with positive mechanical properties and with low price. However, defects such as porosity, which is primarily caused by air entrapment by melt during the filling phase, influences the cast quality [1,2]. Decreased quality caused by air entrapment in the cast volume manifests itself mainly by decline in mechanical properties and machinability. Apart from the aforementioned difficulties, air entrapment leads to reactions of oxygen with chemical components in the melt during mold filling, where it forms oxide inclusions that can be distributed to the cast volume. Regarding aluminum casts, the metal is transferred through free surface turbulences and primarily oxidized skin comes into contact with the melt and other oxides and can form double oxide films—Bifilms—which eventually manifest themselves as channels and reduce resistance of casts against mechanical stress [3,4,5,6].

Currently, the factor being discussed most and which influences reduction and pressing of pores, is holding pressure. It has been proved that increased values of holding pressure positively influence distribution, size and volume of pores as well as cast tightness. When the gate gets solidified, the volume of pores slightly increases which is caused by the melt shrinkage during solidification. Holding pressure can reduce porosity during the final phase of the die casting cycle yet it cannot prevent air entrapment by the melt when it transfers through runners and the mold shaping cavity [7,8,9].

From the technological point of view entrapment of air and gases in the melt can be decreased by suitable setting of input parameters of die casting cycle. Input technological parameters influence size and distribution of pores in the cast volume and filling mode of mold shaping cavity. Filling mode depends on velocity of melt flow when transferring the gate. The velocity is directly proportional to the velocity of a piston in the filling chamber. Several scientific theses have proved correlation of pressing velocity and porosity of casts or of air entrapment in the filling phase. Higher pressing velocity changes character of the melt in the runner from laminar—Planar to turbulent—Non-planar which causes discontinuity of the melt flow. Decreasing of pressing velocity can cause the melt flow to become calmer by means of which continual and regular face of the melt flow can be reached along the entire cross-section of the gate not entrapping the gas or air in its volume. On the other hand, in this it is assumed that prolongation of the casting cycle at low pressing velocity results in reduction of the melt temperature due to long period of duration of die casting cycle which leads to other defects such as cold laps and weld lines [3,10,11,12].

Regardless of technological parameters the key factor influencing the air entrapment in the melt volume is a correct structure of both the gating and the venting system of the mold [12,13,14]. Reaching the high quality of die casting depends on the filling mechanism of the mold cavity by the molten metal. The filling mode is influenced by a number of factors including the shape and the capacity of the cast, weight ratio between the cast and the mold, arrangement of the gating system, shape and area of the gate, volume of mold cavity, area and arrangement of venting holes [13,15]. It has been proved that approximately 90% of defects occurring in the components produced in the die casting process are caused by the errors of the mold design [16,17,18].

To produce high-quality casts it is very important to monitor the process of mold cavity filling and to predict the air entrapment. The filling process can be improved and the extent of defects can be reduced by the change of gating system and venting system on the level of the mold and on the basis of information about the melt flow gained through the monitoring in the mold. For visualization of the mold cavity filling and for prediction of cast defects it is suitable to use numerical modelling and simulation programs. Numerical modelling and simulation of the mold filling process offers considerable advantages in improving the practical technologies in fast prototype processes such as high pressure die casting. Computer supported technology of casting can, during the shortest time possible and without conventional experiments and errors in the foundry plant, optimize the process in the project phase of production which means that sources (material, time and money saving) inevitable for experiments and optimization of the process are minimized. It has been proven that the use of Computer Aided Engineering (CAE) support in the foundry practice for simulation of process occurring during die casting saves 40% of time inevitable for the cast design, 30% of time inevitable for verification of results in laboratories and brings the yield increased by 25% [17,18,19,20,21].

Publication [22] solves the influence of technological parameters of casting on the quality of casts. Parameters considered to be significant and influencing the cast quality include the piston velocity and holding pressure. Publication [23] solves the influence of structural adjustment of the gating system on quality of casts. The gate is defined as determining structural node of the gating system influencing the quality of casts. Right in the gate final acceleration of the melt flow occurs as well as its forming before entering the mold shaping cavity. Velocity in the gate determines the filling mode of the mold shaping cavity on the basis of which it is possible to predict development of the gas entrapment in the cast volume [24,25,26,27,28].

The paper is devoted to evaluation of suitability of change of input parameters of casting contrary to change of structure of the gating system. The objective pursued is to achieve reduction of distribution of gases to the cast volume during die casting cycle. On the basis of knowledge and information published in [10,11,12,13,14,15,16,17,22,23,24,25,26,27,28] it is possible to understand the system of filling chamber—shaping cavity—as a closed system (pipeline). On the basis of hydrodynamics it can be thus assumed that velocity in the gate with preservation of constant flow rate of the melt is influenced by the cross section of the gate or by velocity of the piston. As velocity of the piston represents the input technological parameter of die casting cycle and cross section of the gate is the structural parameter of the gating system, velocity of the melt flow can be thus understood as a correlation factor between technological parameters and gating system structure. By means of simulation program MAGMASOFT 5.3 it was possible to perform the tests of the gating systems with variable height of the gating system b_n_ in case of which the change of average velocity in the gate v_G_ was monitored with preservation of constant velocity of the piston v_P_. On the basis of the detected values of average velocity v_G_ in the gate it was possible to determine the variable velocity of the piston v_Pn_, with the use of continuity equation. By means of these values of variable velocity simulated was the melt flow through the gating system with preservation of constant cross section of the gate S_G_. It was found out that the increase of velocity of the piston v_P_ supports distribution of gases to the cast volume. Consequently, the cause of increase to fair entrapment in the cast volume was solved. Five sets of casts were produced with preservation of constant parameters of casting, with variables b_n_ and v_Pn_ in case of which metallographic evaluation of inner homogeneity of samples taken from the main runner was carried out. It has been proven that increase of velocity of the piston causes increase of velocity in the gate, which is accompanied by the melt whirling and entrapping of gases into its volume. These are subsequently distributed to the mold shaping cavity and remain closed in the cast volume. As long as the melt velocity in the gate v_G_ is considered to be the determining factor of the filling mode of mold shaping cavity, it was proved on the basis of performed measurements and experiments that the change of gating system structure is more advantageous with regards to inner soundness of the cast than the change of setting of technological parameters.

## 2. Materials and Methods

Experimental study was realized regarding gating systems appertaining to the cast of a flange of electromotor which is cast from alloy EN AC 47100 (AlSi12Cu1(Fe)). Numerical simulation of fair entrapment in the cast volume is evaluated in the points in which further mechanical machining of casts is carried out (Figure 1). In these points during filling of mold shaping cavity, the cores forming structural holes in the cast are bypassed. During bypassing of the cores, confluence of two flows of the melt occurs and thus a precondition of air entrapment in the cast volume is formed. Measurement points are in the distance of 3 mm behind the core and 2 mm far from the cast volume into its volume (C1a–C4e).

Experimental study of the air entrapment included five alternatives of the gating system with the variable height of the gate b_n_. The structure of the gate is based on the methodology described in the publication [23]. The gate length for the particular type of the cast is constant according to the aforementioned publication and methodology of structural solution of attachment of the gate to the cast with cylindrical area. From the point of view of structure, the sole parameter influencing the method of filling of the mold shaping cavity is the gate area S_G_. As the gate area is in this case the function of width and height of the gate and width is constant according to [23], the only variable parameter is the gate height b_n_ (Figure 1) [24]. Table 1 presents the evaluated constant and variable structural parameters of the gate. The upper limit of the gate height b_1_ is selected on the basis of calculation according to [23]. The lower limit of the gate height b_5_ is determined on the basis of minimal tolerated value of the gate height defined by the ČSN 22 8601 (in the Czech Republic the standard is given without the ISO equivalent) [29]. The mean values of height of the gate b_2-4_ are selected by the experiment.

The gating system with the gate height of b_1_ = 1.25 mm was selected as the reference sample. As the performed experiments evaluate the influence of the gate height contrary to the influence of the piston on gas entrapment in the cast volume, it was necessary to select correct velocity of the piston. Piston velocity v_P_ for the reference gating system with the gate height b_1_ was determined on the basis of diagram according to Figure 2 [30]. As the mean thickness of the cast h reaches the value of 2 mm for the particular cast and maximal cast length l (in this case it is the cast diameter) reaches the value of 116.5 mm, the melt velocity in the gate for reference gating system is determined to the value of v_G1_ = 35 m·s^−1^. By means of continuity Equation (1) and taking into consideration the closed system [10,11,12,13,14,15,16,17,22,23,24,25,26,27,28], piston velocity is determined v_P1_ = 2.8 m·s^−1^:S_G_ × v_G_ = S_P_ × v_P_,(1)

-S_G_—ingate area (m^2^)-v_G_—melt velocity in the ingate (m·s^−1^)-S_P_—area of piston (0.0038465 m^2^)-v_P_—piston velocity (m·s^−1^)

Air entrapment in the cast volume in the measurement points C1a–C4e (Figure 1) was studied by simulation program Magmasoft MAGMA 5.3–HPDC module, Results–Air Entrapment. Setting of input parameters of die casting cycle is given in Table 2. 

As it is mentioned by Bi, C. et al. [15], to improve accuracy of simulation and to obtain better description of the target entity, it is always advantageous to use networks with higher fineness and high efficiency of generation. Therefore, a fine network with a number of cells amounting to 52,635,960 was selected in simulations and the gating system itself consists of 1,640,521 cells. The network in casts is generated by the components with the size of 0.66(X) × 0.33(Y) × 0.66(Z) mm and the network in the gates is generated by the components with the size of 0.55(X) × 0.25(Y) × 0.55(Z) mm.

After realization of simulations in the case of the gating system with variable height of the gate, on the basis of which average velocity of the melt was determined in the gate v_G_, the equation of continuity (1) was used to determine the values of piston velocity v_P_, as technological equivalents in relation to gate structure. Values of piston velocity v_P_ are given in Table 2.

When comparing the values of the melt velocity in the gate v_G_ and of the velocity in the gate measured after change of technological parameter, i.e., after change of control velocity v_GC,_ observed can be evident deviation Δv_G_ = 6.4%. Deviation of velocity can be given by different shape velocity profile of the melt in the cross-section of the runner. The equation of continuity takes into consideration, in its simplified entity, average velocity of the flow which is in fact a parameter difficult to be detected and maintained [31,32].

The study of the influence affecting distribution of gases to the cast volume involves two assumptions as follows:(a)If technological parameters of die casting are preserved and variable structural elements determining the melt velocity in the gate is height of the gate, and distribution and entrapment of gases in the cast volume is influenced only by formation of the melt flow during transfer through the gate, i.e., by the melt velocity in the gate. The character of the melt flow in the runners remains constant.(b)If the structure of the gating system remains unchanged and the variable parameter determining velocity in the gate is the piston velocity, distribution of gases and entrapment of gases in the cast volume is influenced not only by formation of the melt flow during transfer through the gate but by the flow character in the runners as well. The melt flow in the runners has variable character.

Verification of the aforementioned preconditions is realized by means of cooperation between simulation of casting process and evaluation of the inner porosity of samples taken from the main runner, according to Figure 3. 

On the basis of values of piston velocity obtained by means of simulations and numerical calculations, the casts with variable height of the gate were produced with technological settings identical to the settings in simulations (Table 2). Consequently, with constant height of the gate b_1_ = 1.25 mm, the sets of casts with variable velocity of the piston were produced. The machine Müller Weingarten 600 was used for realization of experiments.

Analysis of porosity of metallographic specimen was carried out by means of microscope Olympus GX51 with 100 times magnification. The results were processed by the program ImageJ which evaluated percentage of porosity in the examined point.

With the use of simulation program Magmasoft MAGMA 5.3–HPDC module, Results–Velocity was in selected points carried out, according to Figure 1, evaluation of the melt velocity in the runners. By means of compilation of the values of the melt velocity in the runners obtained through simulation and of the values obtained by porosity analysis, the mode of the melt flow in the runner and entrapment of gasses in its volume was clarified.

## 3. Results

Results obtained by means of experiments can be divided into three parts. The first part of the results represents the value of gas entrapment in the cast volume immediately before the completion of the filling phase. Consequently, the velocity was evaluated along with the filling mode in the runners for the individual alternatives of gating systems and settings of pressing velocity. Finally, homogeneity of testing samples taken from the runners was examined.

### 3.1. Gas Entrapment in the Cast Volume

Gas entrapment was evaluated immediately before completion of the filling phase when the entire gating system including overflows was 100% full, right before the holding pressure phase was triggered. This interval was selected with regards to the fact that holding pressure positively influences distribution, size and volume of pores. When the cast has been solidified, the volume of pores slightly increases, which is caused by shrinkage of the melt during solidification [8,33]. Therefore, it is relevant to evaluate gas entrapment in the cast volume immediately before triggering of the holding pressure phase. 

Table 3 presents achieved results during examination of gas entrapment in the cast volume for the individual alternatives of the examined gating systems.

A better visual idea of development of gas entrapment in the cast volume in dependence on monitored parameters is offered by graph in Figure 4.

As it becomes evident according to Figure 4, reducing cross-section of the runner and increasing velocity of pressing piston resulted in increase of gas entrapment in the cast. As the velocity in the gate is relatively on the constant level for the individual pairing parameters (filling mode of the mold shaping cavity determines the melt velocity in the gate [22,23,24,25,26,27,28,33]), it is inevitable to focus on evaluation of the flow in the runners.

### 3.2. Evaluation of Flow in the Runners

The runner flow mode is determined by the melt velocity. Increasing melt velocity results in increase of risk of melt splattering when flowing, disturbance of flow continuity and air entrapment in its volume [3,8,9,10,11,12]. Due to the aforementioned, the melt velocity was examined in the runners with the individual examined equivalents of the gating systems. 

#### 3.2.1. Evaluation of the Melt Velocity in the Runners

Table 4 presents average values of the melt velocity measured in the measurement points, as per Figure 1. Average velocity in the runners was defined as an average of values read from simulation in the individual measurement points. With regards to preservation of filling continuity, the pressure losses and relevance of results taken into consideration were only velocity values in the runners during period when the melt was flowing through the entire cross section of the gate [25], i.e., from the time when 67% of the gating system volume was filled up before the filling phase completion.

#### 3.2.2. Assessment of Flow in the Main Runner

As the average melt velocity in the runners reached relatively constant values with variable height of the runner, examination of melt flow was aimed at gating systems filled at variable velocity of pressing piston. The gating system with gate height of b_1_ = 1.25 mm and with pressing piston velocity of v_P1_ = 2.80 m·s^−1^ was used as the referential.

Figure 5 and Figure 6 show points in case of which the most considerable entrapment of gasses by melt occurs. Figure 5 presents formation of the melt flow during transfer from the biscuit to the main part of the runner, in the area between points of temperature measurement (Figure 1). According to Figure 5, it becomes evident that in case of each equivalent a wave effect occurs and thus a backward wave reflected from the opposite runner is formed. The backward wave entraps the gasses in the melt and supports their distribution to the cast. The heaviest gas volume is entrapped in this section when the pressing velocity is set to the value of v_P5_ = 4.78 m·s^−1^. Such setting of the pressing speed induces the other extreme. Should the effect of “initially solidified crust” prevail at lower speed, the melt flow proceeds along the initially formed crust consisting of melt being solidified during contact with the cold face of the mold. At velocity reaching the value of v_P5_ = 4.78 m·s^−1^ the crust gets torn off and has the character of random spattering, which results in larger free surface of the melt coming into the contact with gasses in the mold cavity (Figure 5 and Figure 6). It creates conditions supporting the melt oxidation and gas entrapment in its volume [3,4,34].

Figure 6 presents gas entrapment by the melt in the area of branching of the runner. According to the simulation it becomes evident that in this point the wave effect can be observed along with gas entrapment in the wave. As it has already been mentioned, when the piston velocity reaches the value of v_P5_ = 4.78 m·s^−1^ the melt gets split in the runner and the metal flow does not reach the area of runner branching as the whole. Such situation allows formation of a larger area for the gas entrapment in the melt (Figure 6). When the melt is entrapped and secondary runners get filled at lower speed values, the variant with piston velocity of v_P5_ = 4.78 m·s^−1^ creates in the point of branching “hollow“ spots with gas content (Figure 6).

For completeness it must be mentioned that piston velocity variant of v_P4_ = 4.34 m·s^−1^ had temporary character with flowing elements of lower velocity with implications of melt spattering and flow face branching.

### 3.3. Evaluation of Homogeneity of Testing Samples

Microscopic analysis evaluated homogeneity of samples taken from the main runner according to Figure 3. As the average melt velocity in the runner reached the variable height of the runner, relatively constant values, macroscopic study of homogeneity was realized with the samples taken from the gating systems filled at variable velocity of the pressing piston. The gating system with gate height of b_1_ = 1.25 mm and with pressing piston velocity of v_P1_ = 2.80 m·s^−1^ was used as the reference. 

The measured values of porosity shown in Table 5 were evaluated by the program of ImageJ. Percentage of sample porosity is evaluated as portion of porosity in the area of scratch pattern. 

Figure 7, Figure 8, Figure 9, Figure 10 and Figure 11 present microstructures of the individual analyzed samples (Representative samples selected from a set of test samples) along with evaluation of porosity of by means of the program ImageJ.

According to Table 5 and Figure 8, Figure 9, Figure 10 and Figure 11 (Representative samples selected from a set of test samples) it is clear that velocity of pressing piston increases along with percentage of porosity in scratch pattern of samples. With regards to results presented in Table 5 it can be stated that increasing velocity of pressing piston changes the character of melt flow in the runners which supports air entrapment in the melt volume.

## 4. Discussion

According to the results of the experiments (Table 4, Figure 4), the structure change along with the change of setting of technological parameters directly influence values of gas entrapment in the cast volume.

### 4.1. Evaluation of Gas Entrapment in the Cast Volume in Dependence on Change of the Ingate Height b_n_

Figure 4 shows that gas entrapment in the cast volume is more noticeable in the case of pressing velocity increase, contrary to change of the gate height. According to the aforementioned facts [24,25,26,27,28] it is clear that the area of the gate and the melt velocity in the gate determine the filling mode of mold shaping cavity which allows prediction of development of gas entrapment in the cast volume. Decreasing height of the gate b resulted in increase of percentage of air entrapment in the cast volume. Table 3 offers explanation for the aforementioned facts. Smaller area of the gate allows the increase of melt velocity in the gate which shifts filling mode of mold shaping cavity from the turbulent to the disperse one [19,20,24,25].

Referring to Table 4, which presents the values of the melt velocity in the runners, it can be stated that during change of the gate structure the value of air entrapment in the cast volume is influenced only by the structure of the gate and by the melt velocity in the gate. Slight deviations of the measured velocity values of the melt in the runners can be explained by differentiation of the velocity profile in the piping cross [32,33,35].

The aforementioned proves the following hypothesis: “(a) If the technological parameters remain unchanged and the gate height represents a variable structural element determining the melt velocity in the gate, distribution and entrapment of gasses in the cast volume is influenced only by formation of the melt flow when transferring through the gate, i.e., by the melt velocity in the gate. The melt flow character in the runners remains unchanged.”

### 4.2. Evaluation of Gas Entrapment in the Cast Volume in Dependence on Change of the Pressing Pistion Velocity v_P_

Even though melt velocity in the gate selected as correlation factor remains on a relatively constant level (deviations are explained in [31,32]), on the basis of measurements presented in Figure 4 and Table 3, the values of air entrapment in the cast volume show higher values in case of change of pressing velocity than during the change of structural solution of the gate. It is clear that the gas entrapment in the cast volume increases along with pressing piston velocity.

The explanation can be derived from the measurements presented in Table 5. Pressing piston velocity increases along with the velocity value in the runners which results in change of mode of the melt flow in the runners.

Assumption of the flow mode change is proved on the basis of Figure 5 and Figure 6. The wave effect can be observed in the monitored points, where the wave is formed as the result of the influence of the melt impact against the opposite wall of the runner under which the gas entrapment in the melt volume along with its distribution to the cast volume occurs. Increasing pressing piston velocity makes the effect stronger. At velocity of v_P5_ = 4.78 m·s^−1^ not only whirling of the melt caused by the wave formation during impact against runner’s wall can be observed, but also strong melt spattering might be noticed as well through which larger area of the melt is created that allows gas entrapment and oxidation of the melt which gets into contact with gasses. On the basis of simulations it was proved that the face of the melt flow at pressing piston velocity of v_P1_-v_P3_ remained unchanged without a change in direction during transfer through the straight runner and proceeded according to the model of “initially solidified crust”. In case of v_P5_ the face of the melt flow was not compact, showing marks of random spattering. At pressing piston velocity of v_P4_ the melt flow had character of the straight face of the flow and showed the marks of only slight “spattering” and splitting. 

When gas entrapment during transfer of the melt through the runner was verified, the scratch patterns of the samples taken from the runners were realized (Figure 3). The evaluated parameter was the porosity volume in the scratch pattern area.

According to Table 5 and Figure 7, Figure 8, Figure 9, Figure 10 and Figure 11, porosity in scratch patterns of the samples increased along with pressing piston velocity. The fact proves increased distribution of gasses to the cast volume and explains values of gas entrapment in the cast volume according to Table 3. 

The following hypothesis was proved by the experiments and obtained results: “(b) if the gating system structure remains unchanged and variable parameter determining velocity in the gate is pressing piston velocity, distribution and entrapment of gasses in the cast volume is influenced not only by formation of the melt flow during transfer through the in gate, but also by the character of the flow in the runners. The melt flow in the runners has variable character.”

## 5. Conclusions

The paper deals with the issue of advantages of change in setting of technological parameter in comparison to change in structure of gating system with regards to inner soundness of the cast. Correlation factor determining the filling mode of mold shaping cavity is selected melt velocity in the runner. The obtained results have proved that the change in structure of the gating system or of its structural node is with regards to gas entrapment in the cast volume, more advantageous than the change in setting of technological parameters of the pressing system.

It has also been proven that change in structure of the gate influences the change of the melt velocity in the gate along with the filling mode of the mold shaping cavity. Diminishing of the gate area results in increase of gas entrapment in the cast volume which is caused by the shift of filling mode from the turbulent to the disperse character. The melt flow mode in the runners is not subjected to the influence of the change in the gate structure and thus it can be claimed that gas entrapment during the melt flow through the runners has constant character.

Contrary to the aforementioned, change of the pressing piston velocity influences series of interrelated parameters. Pressing piston velocity increases along with the melt velocity in the runners which consequently changes the melt flow mode. Change of the flow mode in the runners, which brings along splitting of the straight face of the flow, whirling and spattering of the melt, supports gas entrapment in the melt volume and melt oxidation when its free surface gets into contact gasses and distribution of gasses to the cast volume. 

On the basis of the obtained results the following recommendations can be proposed:introduction of new cast production should be accompanied with paying high attention to design of the gating systems and of their structural elements,of gas entrapment by the melt in initial design of the gating system structure,correct design of the gating system shall result in more stable casting cycle with the possibility of partial regulation by means of change of technological parameters,in introduction of new casts into production it is desired to use CAx systems which allow detecting hidden faults of structure and technology design.

Future research will be focused on the prediction of castings mechanical properties. The aim is to follow the creation of a mathematical model and determining properties of suitable ratio between construction of inlet system and setting the casting technological parameters.

## Figures and Tables

**Figure 1 materials-14-02264-f001:**
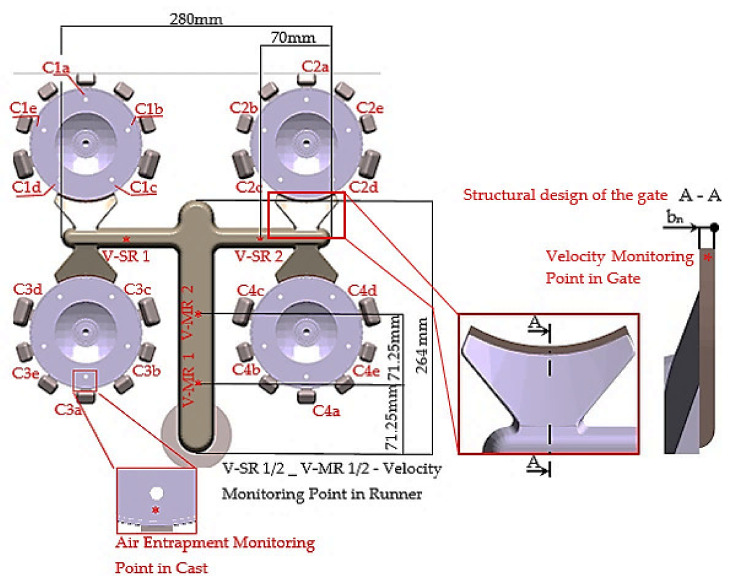
Location of measurement points.

**Figure 2 materials-14-02264-f002:**
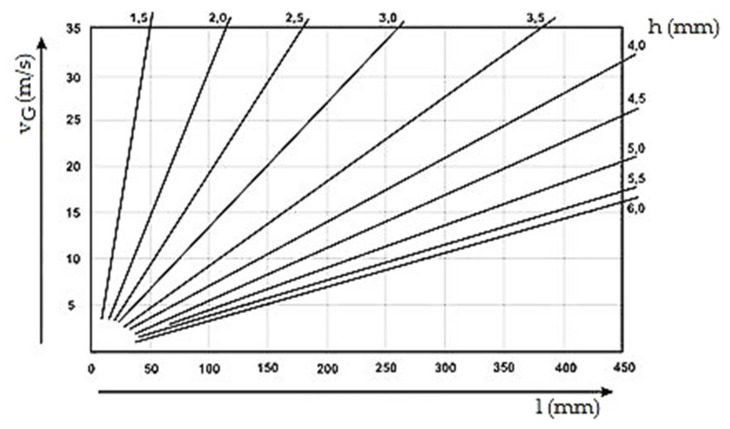
Speed of flow v_G_ inside the gate (Gate velocity) dependence on the wall thickness h of a cast and in dependence of maximum distance l mold on gate.

**Figure 3 materials-14-02264-f003:**
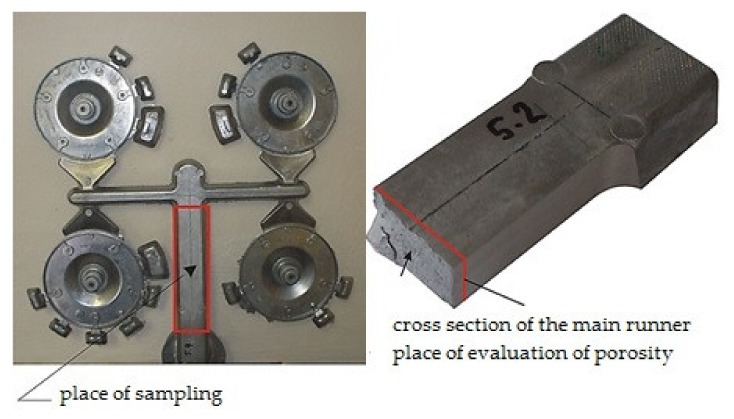
The sampling point for porosity analysis.

**Figure 4 materials-14-02264-f004:**
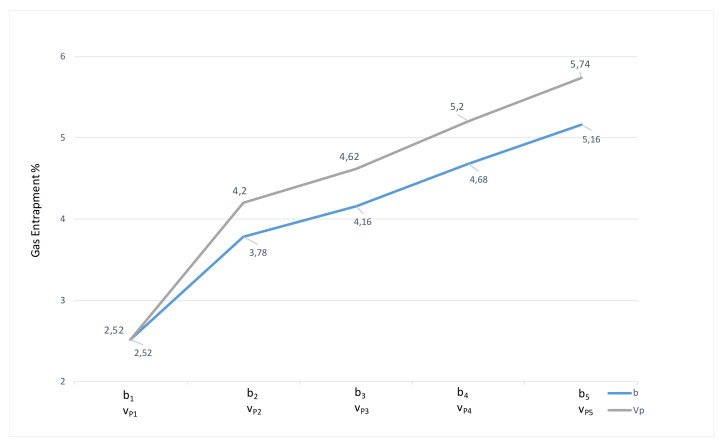
Gas entrapment in the cast in dependence on monitored parameters.

**Figure 5 materials-14-02264-f005:**
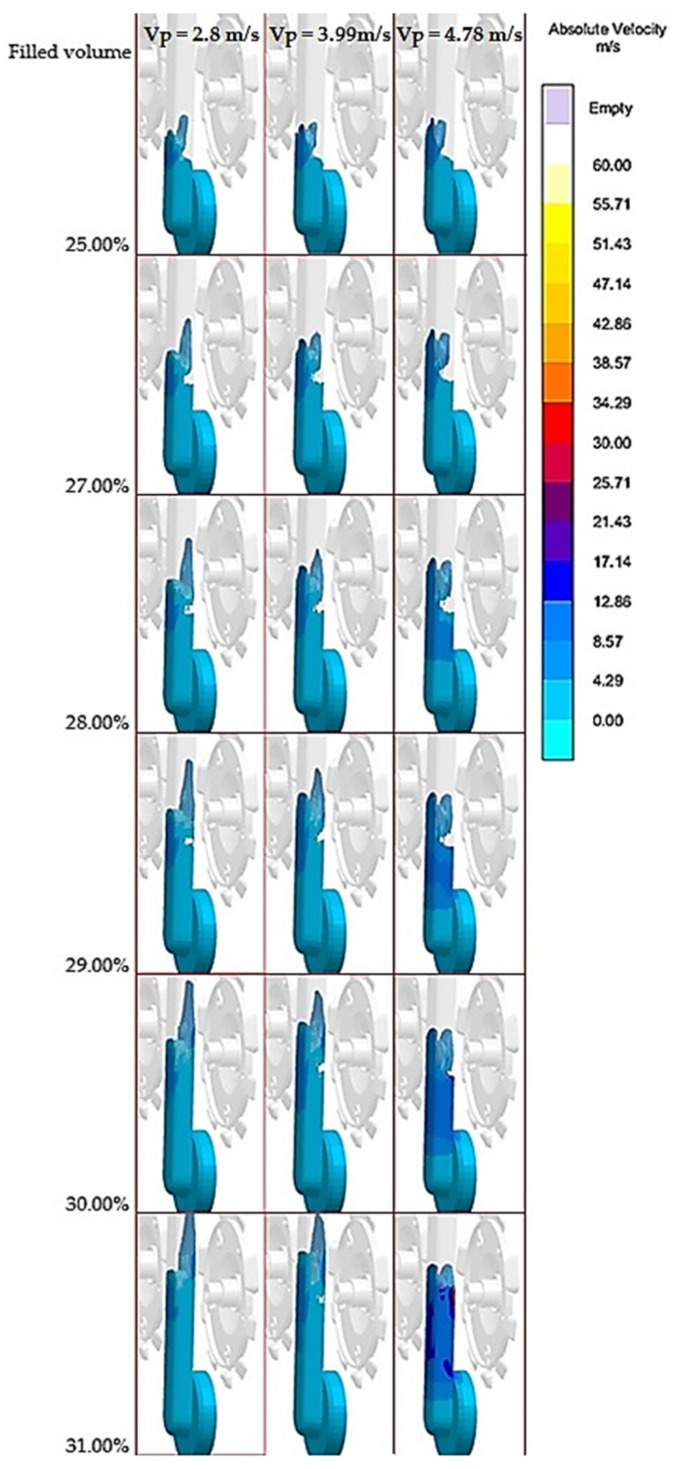
Gas entrapment by the melt in the runner in the area of measurement points of velocity.

**Figure 6 materials-14-02264-f006:**
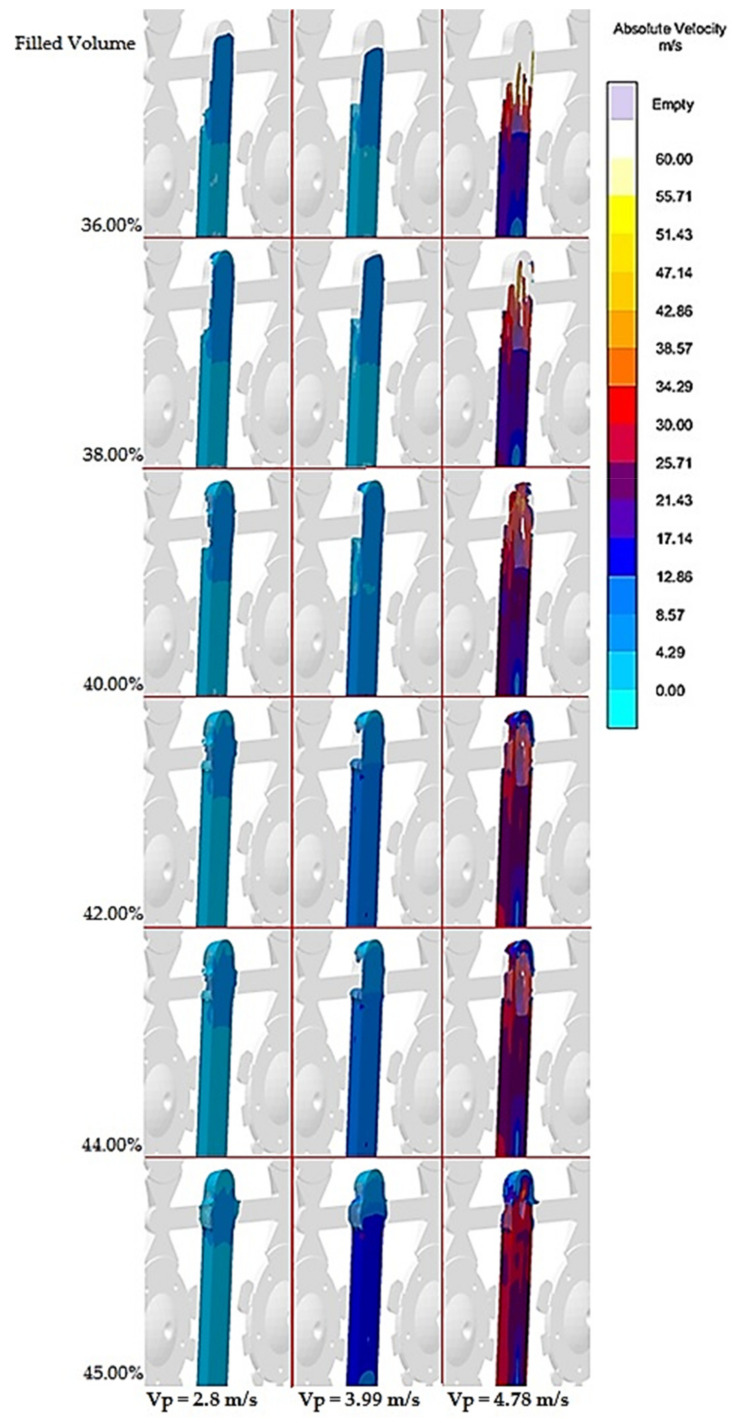
Gas entrapment by the melt in the runner in the area of runner branching.

**Figure 7 materials-14-02264-f007:**
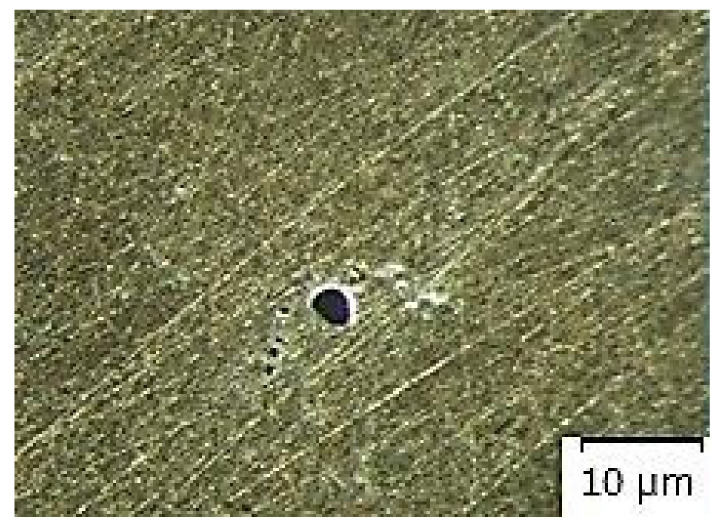
Microscopic evaluation of sample 1/×100/.

**Figure 8 materials-14-02264-f008:**
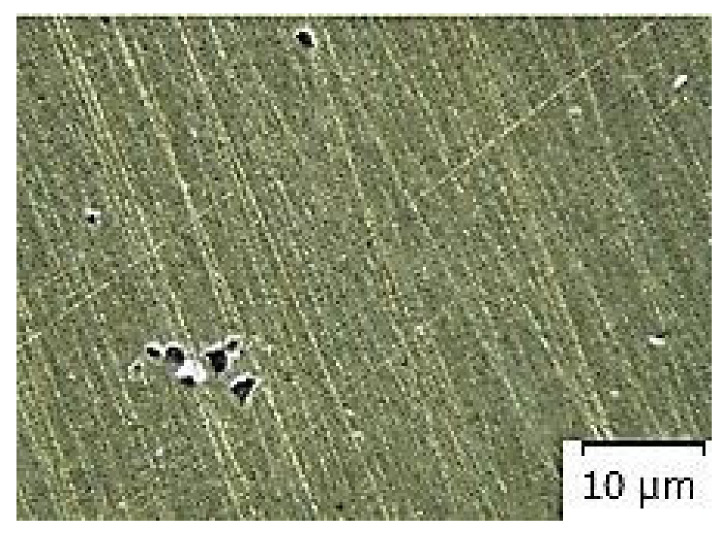
Microscopic evaluation of sample 2/×100/.

**Figure 9 materials-14-02264-f009:**
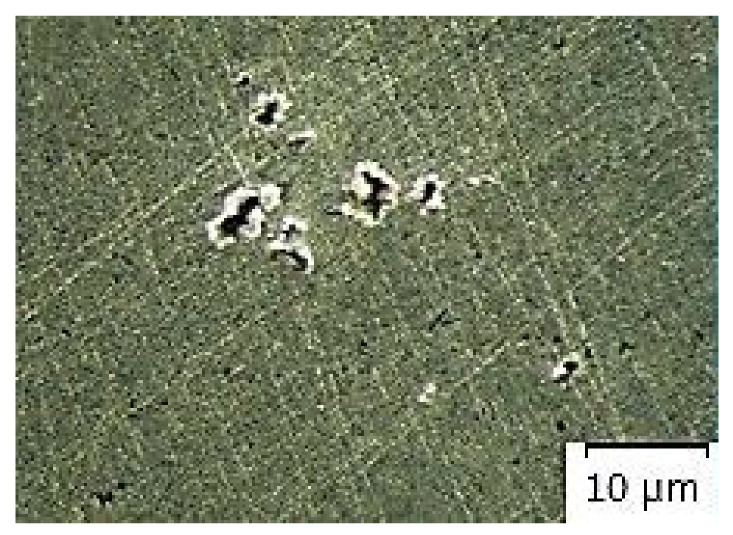
Microscopic evaluation of sample 3/×100/.

**Figure 10 materials-14-02264-f010:**
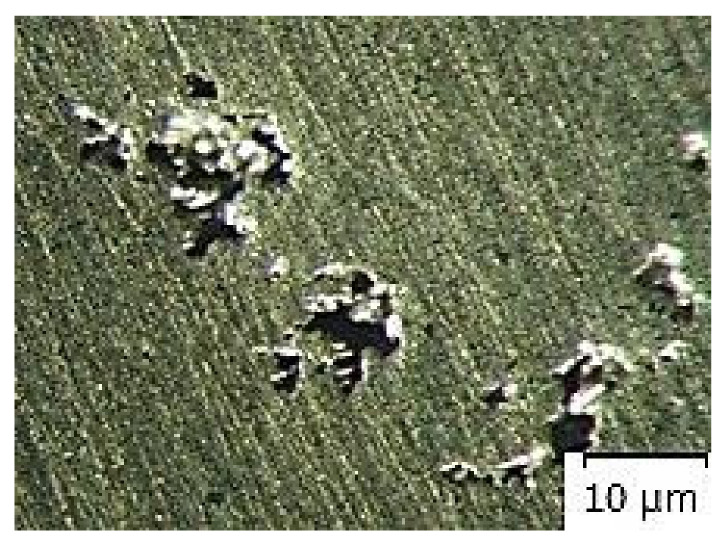
Microscopic evaluation of sample 4/×100/.

**Figure 11 materials-14-02264-f011:**
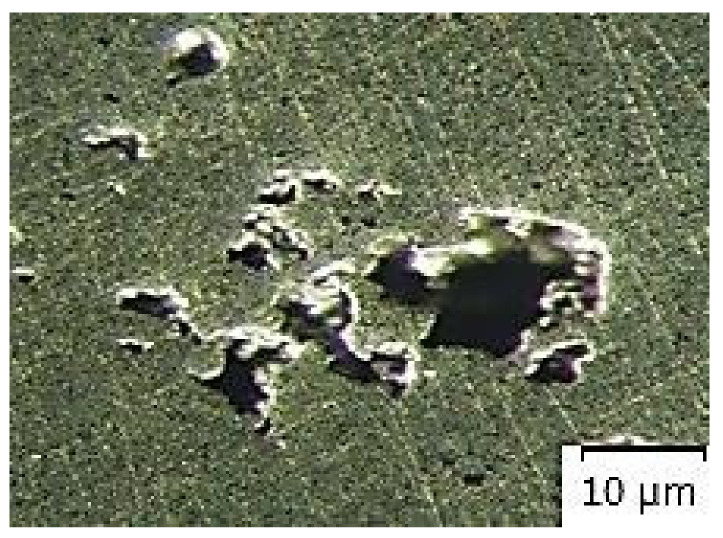
Microscopic evaluation of sample 5/×100/.

**Table 1 materials-14-02264-t001:** Gate dimensions.

Gate Height b (mm)	b_1_	b_2_	b_3_	b_4_	b_5_
	1.25	1.03	0.92	0.82	0.75
Gate Area S_G_ (mm^2^)	S_G1_	S_G2_	S_G3_	S_G4_	S_G5_
	76.210	62.797	56.090	49.994	45.726
Gate Length a (mm)	60.968				

**Table 2 materials-14-02264-t002:** Determination of piston velocity with the use of correlation factor.

Structural Element	Correlation Factor	Technological Factor	Melt Velocity in the Gate Measured during Change of Setting–Control v_GC_, m·s^−1^
Gate Height b, mm	Melt Velocity in the Gate v_G_, m·s^−1^	Piston Velocity v_P_, m·s^−1^	
b_1_ = 1.25	v_G1_ = 35.00	v_P1_ = 2.80	v_GC1_ = 37.89
b_2_ = 1.03	v_G2_ = 47.44	v_P2_ = 3.76	v_GC2_ = 50.41
b_3_ = 0.92	v_G3_ = 50.43	v_P3_ = 3.99	v_GC3_ = 53.46
b_4_ = 0.82	v_G4_ = 54.80	v_P4_ = 4.34	v_GC4_ = 58.10
b_5_ = 0.75	v_G5_ = 60.33	v_P5_ = 4.78	v_GC5_ = 63.95

**Table 3 materials-14-02264-t003:** Gas entrapment in the cast volume.

Constant Piston Velocity v_P1_ = 2.80 m·s^−1^		Constant Gate Height b_1_ = 1.25 mm	
Gate Height (mm)	Gas Entrapment (%)	Piston Velocity (m·s^−1^)	Gas Entrapment (%)
b_1_ = 1.25	2.52	v_P1_ = 2.80	2.52
b_2_ = 1.03	3.78	v_P2_ = 3.76	4.20
b_3_ = 0.92	4.16	v_P3_ = 3.99	4.62
b_4_ = 0.82	4.68	v_P4_ = 4.34	5.02
b_5_ = 0.85	5.16	v_P5_ = 4.78	5.74

**Table 4 materials-14-02264-t004:** Melt velocity in the runners.

Constant Piston Velocity v_P1_ = 2.80 m·s^−1^			Constant Gate Height b_1_ = 1.25 mm		
Gate Height (mm)	Average Velocity in Main Runner (m·s^−1^)	Average Velocity in Secondary Runner (m·s^−1^)	Piston Velocity (m·s^−1^)	Average Velocity in Main Runner (m·s^−1^)	Average Velocity in Secondary Runner (m·s^−1^)
b_1_ = 1.25	14.36	24.08	v_P1_ = 2.80	14.36	24.08
b_2_ = 1.03	15.09	24.10	v_P2_ = 3.76	21.53	32.60
b_3_ = 0.92	14.59	23.54	v_P3_ = 3.99	24.47	34.70
b_4_ = 0.82	14.68	23.96	v_P4_ = 4.34	27.84	36.09
b_5_ = 0.85	14.48	23.13	v_P5_ = 4.78	29.34	39.35

**Table 5 materials-14-02264-t005:** Average porosity values of samples taken from the main runner.

Sample No.	Piston Velocity (m·s^−1^)	Porosity f, (%)
1	v_P1_ = 2.80	0.27
2	v_P2_ = 3.76	0.33
3	v_P3_ = 3.99	0.65
4	v_P4_ = 4.34	3.14
5	v_P5_ = 4.78	5.85

## Data Availability

Data sharing is not applicable to this article.

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
