# Peer review of "Research and Evaluation of the Influence of the Construction of the Gate and the Influence of the Piston Velocity on the Distribution of Gases into the Volume of the Casting"

_materials, 2021, doi:10.3390/ma14092264_

Round 1

Reviewer 1 Report

Dear Authors,
The paper sent to review will be quite interested for foundries plants employees. The Authors applied the most popular and recognized as the best simulation code (system) named MAGMA. The reviewed paper should be considered as being at average level. Title and keywords are appropriate and adequate to paper content. The results are presented comprehensively. The literature review is up-to-date. The goals of the paper are sufficiently explained. The conclusions are adequate.
There were found some issues which require additional correction. They were listed below:
- line 17 page 1, l.1 p. 3, l. 408 p. 14 - "The submitted article..." - should be In the paper... -the word submitted refers to not published works,
- add in abstract what is the novelty of your subject,
- l.34 p.1 - is "...influence..." should be ...influences...
- l.34 and l.40 p.1 - is "oxidic" more suitable would be oxide
- l.83 p.2 - is "...high pressure die casting (HPCD)..." this abbreviation was explained in l.31 p.1; moreover more proper is high-pressure die-casting
- l.87 p.2 - is "...CAE..." explain this abbreviation, please - Computer Aided Engineering
- l.167-171 p.4 - correct the powers of units e.g. m2 etc.
- Figure 3 p.6 - in the figure one can see that not each chill-vent was filled. Why? It is known that proper filling of vents fulfil an assumed quality of process. Moreover to the tests the Authors have chosen the plane casting, which in HPDC process has low porosity contribution comparing the castings with diversify wall thickness. It should be mentioned here that the Authors of this paper analyze the discontinuities in the gating system. However, they indicate in the last paragraph of the conclusions that further work will concern the study of mechanical properties. Thus, the shape of the casting will have a high influence here,
- l.243 p.7 - is "...the volume of pores slightly increases which is caused by shrinkage..." what is the contribution of shrinkage porosity? The term "slightly" is to qualitative,
- p.7 Table 3 - Split the data, please. Gate height, mm; Gas Entrapement, % and Piston Velocity, ms-1; Gas Entrapement, %. It makes confusion. It should be stated in the description of second part that the constant value for b=1.25mm,
- p.8 Table 4 - Split the data, please. According to the example above,
- p.8 l.283 - is "...the tablet to main the runner..." proper name on tablet is biscuit. Moreover "...to the main runner...",
- p.10 l.308 - what means the word firma?
- p.11 l.324 and subtitles of Figures 7-11 Author write about "Macroscopic evaluation.../x100" While the macroscopic examination consists in observing with the naked eye or with a magnifying glass with a magnification not exceeding 30x of surface of objects or sections. Moreover nowadays, the magnification x100, x200 etc. is no longer indicated under the figures. Only the micrograph has a scale, which the authors introduced,
- Figures 9-11 (Figure 8 probably) - the micrographs presents the typical shrinkage porosity. This paper concerns air entrapment discontinuities formation. More examples of air entrapment are needed,
- p.13 l.397-400 - improve this paragraph taking in to consideration the point above,
- p.13 l.401 - correct "...b)...", please,
- p.14 l.436 - correct "...CA-x" to ...CAx, please

The Authors didn't mention that the perfect way to reduce the air entrapment is to apply the vacuum system.

Best regards

Author Response

Dear Reviewer,

First of all, thank you for your time and effort for revising the paper. We have considered the suggestions and comments in the paper. The modifications are highlighted in the revised version of the manuscript. The attached documen provides a point-by-point reply to your comments.

Reviewer 2 Report

This is an interesting high quality paper, just minor improvement suggested:

  1. The abstract contains a symbol vG but not necessary.
  2. The first two paragraph contains more repetition, these can be shortened.
  3. In Introduction bn(1-5) appeared before the meaning of (1-5) introduced. Please in this section just use simply bn, vPn. After the mention the five different experiment set-up, the usege of this symbols will be clear.
  4. The Results part conains new descriptions and citations. These would have a better place in Introduction. But in this structure the article also remains clear.
  5. Table 4. in this form is not so clear as the other part of the article. Please add more detailed description of the contents of table 4. in 3.2.1 section.

Round 2

Reviewer 1 Report

Dear Authors,
All the adjustments in the paper under review, were made. The answers are satisfied for me. Therefore, the paper can be published at present form.
Best regards